# Comparison of Blood and Blood Product Transfusion in COVID-19 and Non-COVID-19 Patients Requiring Extracorporeal Membrane Oxygenation for Severe Respiratory Failure

**DOI:** 10.3390/jcm12144667

**Published:** 2023-07-13

**Authors:** Malindra C. Fernando, Tim Hayes, Martin Besser, Florian Falter

**Affiliations:** 1Department of Anaesthesia and Intensive Care, Royal Papworth Hospital, Cambridge CB2 0AY, UK; malindraclivefernando@gmail.com; 2Department of Anaesthesia and Intensive Care, Manchester University Hospitals, Manchester M13 9WL, UK; tim.hayes@mft.nhs.uk; 3Department of Haematology and Blood Transfusion, Cambridge University Hospitals, Cambridge CB2 0QQ, UK; martin.besser@nhs.net

**Keywords:** SARS-COVID-19, ARDS, ECMO, blood transfusion

## Abstract

COVID-19 has resulted in an exponential increase in patients with severe respiratory failure requiring extracorporeal membrane oxygenation (ECMO). Patients on ECMO regularly require high volumes of blood and blood products but, so far, there has been no comparison of transfusion requirements between COVID-19 and non-COVID-19. Using electronic patient records at two major UK ECMO centres, Royal Papworth Hospital and University Hospital South Manchester, we reviewed the transfusion requirements of patients requiring ECMO between January 2019 to December 2021. A total of 271 patients, including 168 COVID-19 patients were available for analysis. Since COVID-19 patients spent almost twice as long on ECMO (27.1 vs. 14.16 days, *p* ≤ 0.0001) we indexed transfusion in both groups to days on ECMO to allow comparison. COVID-19 patients required less red blood cells (RBC) per day (0.408 vs. 0.996, *p* = 0.0005) but more cryoprecipitate transfusions (0.117 vs. 0.106, *p* = 0.022) compared to non-COVID-19 patients. COVID-19 patients had more than double the mortality of non-COVID-19 patients (47% vs. 20.4%, *p* = 0.0001) and those who died during the study period had higher platelet transfusion requirements (*p* = 0.007) than their non-COVID-19 counterparts. Transfusion requirements and coagulopathy differ between COVID-19 and non-COVID-19 patients. The distinctly different transfusion patterns between the two groups remain difficult to interpret, but further investigations may help explain the haematological aspects of severe COVID-19 infection.

## 1. Introduction

Since its first use in 1972 and initial slow uptake extracorporeal membrane oxygenation (ECMO) has been increasingly used in intensive care units (ICU) around the world to rescue patients with severe respiratory failure [1,2]. There has been unprecedented demand for this technology during the H1N1 pandemic 2009–2010 and even more so during the recent COVID-19 pandemic [3,4].

Thanks to big advances in technology, bleeding has become much less of an issue when treating patients with ECMO. However, it is still associated with high use of blood and blood products. Sheer forces within the circuit and blood loss at the cannulation sites with skin and soft tissue breakdown alongside anticoagulant use are the main reasons for increased transfusion requirements [5,6,7]. Although unfractionated heparin remains the most widely used anticoagulant during ECMO, there have been recent advances using other agents, namely bivalirudin and mafamostat mesilate [8,9]. One large case series with central mostly veno-arterial ECMO describes a reduction in transfusion in the first 24 h after ECMO initiation and reduced mortality in the bivalirudin patients compared to those on heparin [8]. These findings were corroborated in later another single-centre series including veno-venous ECMO patients, where patients on bivalirudin received a significantly lower number of blood and blood products and experienced fewer thrombotic complications [10].

Patients critically ill with COVID-19 are known to be in a highly activated, procoagulant state and are prone to thrombotic complications [7]. Anticoagulation requirements for these patients are often higher than for patients with non-COVID-19 lung failure requiring ECMO [11]. Balancing thrombosis alongside haemostasis in the presence of both COVID-19 and ECMO can be a difficult task. It is suggested that the coagulopathy associated with COVID-19 is a combination of low-grade disseminated intravascular coagulopathy (DIC) and localised pulmonary thrombotic microangiopathy [12]. Early on in the pandemic, an intensification of heparin therapy was proposed, especially in critically ill and ventilated patients [13]. This was given additional weight when it became clear that anticoagulant therapy appeared to be associated with better outcomes in severe COVID-19 [14].

Given the more aggressive anticoagulation in patients severely ill with COVID-19, it seems inevitable that ECMO will go hand in hand with increased transfusion requirements. An early single-centre observational study from the UK seemed to corroborate that [15]. The aim of this retrospective review of practice in two major UK ECMO centres is to ascertain the difference in transfusion patterns in patients treated for COVID-19 and non-COVID-19 respiratory failure.

## 2. Materials and Methods

### 2.1. Study Design

This was a retrospective review of transfusion requirements in patients on VV-ECMO in non-COVID-19 and COVID-19 patients. The study obtained IRB approval at both centres (Reference HRA0002) and the need for retrospective patient consent was waived. We collected data from consecutive patients at 2 UK tertiary lung failure referral centres, Royal Papworth Hospital (RPH) in Cambridge and the University Hospital South Manchester (UHSM), from 1 January 2019 to 31 December 2021. Data were collected using the electronic patient record systems in both hospitals (iMDsoft, Dusseldorf, Germany, in Cambridge; Dendrite Clinical Systems, Reading, UK, in Manchester).

### 2.2. Participants

We included all patients over 18 years old that were admitted to both centres and required ECMO for severe respiratory failure. We excluded any patients who suffered vascular complications as a result of cannulation and received blood or blood products as a direct result.

Both centres adhered to the UK nationally agreed upon inclusion/exclusion criteria for respiratory ECMO, which include significant life-limiting comorbidity such as haematological or solid organ malignancy [16].

### 2.3. Outcome Parameters

The primary outcome was the transfusion of blood and blood products, packed red blood cells (RBC), platelets, fresh frozen plasma (FFP), or cryoprecipitate following the establishment of VV-ECMO. Other variables included time spent on VV-ECMO and in-hospital mortality. Patients were not followed up after discharge from the respective tertiary centres.

### 2.4. Clinical Management

Patients were referred to RPH or UHSM via the national ECMO referral system and were accepted for ECMO treatment according to nationally agreed criteria. Following acceptance, patients were retrieved by the accepting centres’ retrieval team. If required, ECMO was initiated at the referring hospital. The majority of patients had their ECMO cannulas in the femoro-jugular configuration, although difficulty with access necessitated some femoro-femoral cannulation.

Heparin was the standard anticoagulation strategy in both institutions. An initial bolus dose of 2500–5000 IU was administered during the insertion of the ECMO cannula. As is protocol in both institutions, all patients on admission to RPH or UHSM obtained a full diagnostic CT scan. If there was no radiological evidence of bleeding, a heparin infusion was started intravenously until a target APR of 1.5–2.3 was achieved in accordance with hospital ECMO guidelines at both hospitals.

### 2.5. Transfusion Management

Transfusion management was the same at both institutions and was in concordance with Extracorporeal Life Support Organisation (ELSO) guidelines. The transfusion thresholds were haemoglobin ≤ 70 g/L, platelet < 50 × 10^9^/L and fibrinogen < 1 g/L^9^. If there was no clinically significant bleeding, lower platelet counts and fibrinogen levels were accepted by treating intensivists. Viscoelastic coagulation testing (ROTEM Sigma, Werfen, Barcelona, Spain, in Cambridge; TEG 6, Haemonetics, Boston, MA, USA, in Manchester) was used alongside laboratory values to guide transfusion requirements.

### 2.6. Statistical Analysis

Descriptive data are presented as number (%), mean (±standard deviation) or median (95% confidence interval) as appropriate. Group comparisons for categorical data were achieved using Chi-square tests. Normally distributed continuous variables were analysed with the Student *t*-test, and nonparametric variables were analysed using the Mann–Whitney Test. A *p* value < 0.05 was considered statistically significant. Statistical analysis was performed with MedCalc software, Version 22 (Ostend, Belgium).

## 3. Results

### 3.1. Baseline Characteristics

In total, 271 patients receiving VV-ECMO in RPH and UHSM were included from January 2019 until December 2021. Patient baseline characteristics are presented in Table 1. A total of 168 patients had COVID-19, and 103 had respiratory failure from other causes. There were significant differences between the two groups in body mass index (BMI), weight, and sex distribution.

### 3.2. Admission Blood Characteristics

There were significant differences in admission blood tests between the two groups (Table 1). The median admission haemoglobin concentration for COVID-19 was 100 (g/L) and 94 (g/L) for non-COVID-19 (*p* = 0.0282). Overall, COVID-19 patients had a higher admission platelet count (100 vs. 94 × 10^9^/L, *p* ≤ 0.0001) and fibrinogen level (5.29 vs. 4.38 g/L, *p* = 0.0071) compared to the non-COVID-19 patients.

### 3.3. Transfusion

In total, 158 (94%) of the COVID-19 patients received an RBC transfusion compared to 98 (95%) in the non-COVID-19 patients. Transfusion rates for platelets (0.111 ± 0.1639 vs. 0.296 ± 0.638 units/day *p* = 0.363) and for FFP (0.101 ± 0.3157 vs. 0.290 ± 0.8003 units/day *p* = 0.976) were different between both patient groups; however, this did not reach statistical significance. In total, 24 (23%) non-COVID-19 patients received cryoprecipitate in comparison to the 59 (35%) COVID-19 patients.

To account for the significant difference in days spent on mechanical support between the two groups, we indexed transfusion to days spent on ECMO. We found an increase in red cell transfusion per day in the non-COVID-19 patients compared to the COVID-19 patients (0.996 ± 2.67 units vs. 0.408 ± 0.5123 units, *p* = 0.0005). In contrast, the per-day transfusion of cryoprecipitate was higher in the COVID-19 group (0.117 ± 0.2394 units vs. 0.106 ± 0.4831 units, *p* = 0.022). There was no difference in per-day transfusion of FFP or platelets. Table 2 summarises the transfusion data.

We next analysed transfusion-related survival patterns in both groups. In the COVID-19 group, there was an increase in per-day platelet transfusion (*p* = 0.007) amongst patients who died during the study period. There was no difference in transfusion of RBC, FFP, and cryoprecipitate between COVID-19 patients alive or dead. In the Non-COVID-19 group, the non-survivors received significantly more per-day transfusions of RBC (*p* = 0.0008), FFP (0.004), and cryoprecipitate (0.001) (Table 3).

The comparison between survivors and non-survivors follows a pattern similar to that of overall transfusion (Table 4). In both groups, non-COVID-19 patients received more RBC transfusion. In non-survivors, non-COVID-19 patients received more FFP, while in those alive, COVID-19 patients received more cryoprecipitate.

### 3.4. Outcomes

Overall, the COVID-19 patients spent almost double the number of days on ECMO than non-COVID-19 patients (27.1 vs. 14.16 days, *p* < 0.0001)). Equally, mortality was more than double in the COVID-19 group (47.6% vs. 20.4%, *p* = 0.0001) (Table 5).

## 4. Discussion

To our knowledge, we present the largest cohort comparing transfusion patterns between COVID-19 and non-COVID-19 patients on VV-ECMO so far. Unsurprisingly, of the 271 patients included over the 3-year period from January 2019 to December 2021, the number of COVID-19 patients is higher than that of non-COVID-19 patients. Overall, the COVID-19 patients were admitted with higher haemoglobin (*p* = 0.03), platelet count (*p* < 0.0001), and total fibrinogen (*p* = 0.0071).

Both sets of patients were anaemic on admission to the ECMO referral centre, the non-COVID-19 patients more so than the COVID-19 patients. Although statistically significant, the difference between the admission Hb—94 v 100 g/L—is not big enough to explain why the non-COVID-19 group received double the amount of per-day RBC transfusions than their COVID-19 counterparts. None of the included patients suffered vascular complications that led to an increased need for red cell transfusion, which leads us to believe that the difference in transfusion is the result of distinctly different disease processes. The total average RBC transfused was 12 units in both the COVID-19 and non-COVID-19 groups. When indexed to days on ECMO, however, we did observe that per-day RBC transfusion was significantly lower in the COVID-19 group, which is surprising in view of the fact that these patients spent significantly longer time on ECMO. Previous studies demonstrated a clear association between time spent on ECMO and transfusion requirement [17,18]. There was no association of RBC transfusion with mortality in the COVID-19 patients. In contrast, non-survivors in the non-COVID-19 group had a significantly higher per-day RBC transfusion rate than survivors, which is keeping with previously published evidence [5,19].

It is unclear why, in our cohort, the COVID-19 patients had a higher overall platelet count since thrombocytopenia has been described as highly prevalent in patients with severe COVID-19 [20]. There was no significant overall or per-day difference in platelet transfusion between the groups. However, there was an association between daily platelet transfusion and mortality in the COVID-19 patients. It is possible that this association is due to a combination of the disease process and the longer time spent on ECMO resulting in greater platelet consumption in the extracorporeal circuit. We did not observe this association in the non-COVID-19 patients. The reason for this is unclear, given that the non-COVID-19 patients were admitted with lower platelet counts and required higher transfusions of RBC and other plasma products. It is possible that the use of platelet concentrates is a surrogate marker for uncontrollable bleeding or acquired von Willebrand disease. Out of guideline use of cryoprecipitate or FFP is highly unusual and does not cluster in an end-of-life scenario, unless there is significantly abnormal blood clotting. None of the included patients underwent any operations while on ECMO, nor was there any significant blood loss from cannulation sites or through the extracorporeal circuit, which makes this observation even more difficult to explain. Interestingly, we saw a similar trend to different platelet transfusion rates in non-COVID-19 non-survivors, although this did not reach statistical significance.

The supranormal levels of fibrinogen on admission to the ECMO centre in both groups are likely to be the result of an acute inflammatory process and the severity of the underlying disease [21,22]. The higher admission fibrinogen in the COVID-19 group did not translate into decreased cryoprecipitate transfusion in these patients. On the contrary, the COVID-19 patients had a higher number of total and daily units of cryoprecipitate transfused than the non-COVID-19 patients. Dynamic changes in fibrinogen have been observed in severe cases of COVID-19 and were mostly associated with disseminated intravascular coagulation (DIC) [23], which we did not observe in any of the included patients. In both groups, non-survivors received more cryoprecipitate transfusions, which reached significance in the COVID-19 patients. This is difficult to reconcile with the fact that non-COVID-19 survivors received significantly more cryoprecipitate than COVID-19 survivors. Cryoprecipitate is a source not only of fibrinogen but also of factor VIII, XIII, and von Willebrand factor. It is intriguing to think that the additional factors could have had additional benefits in COVID survivors. Alternatively, one could speculate that patients with a less intense acute phase response and fibrinogen increase are more likely to have met the threshold for cryoprecipitate transfusion and may thus have had improved outcomes compared to patients with extremely high fibrinogen levels due to an extreme acute phase response.

There was no difference in FFP transfusion between the groups. Similar to RBC transfusion, we did not see any relationship between FFP transfusion and mortality in COVID-19 patients. We did observe an association between per-day FFP transfusion and mortality in the non-COVID-19 group.

In our cohort, mortality was higher in the COVID-19 patients (47.6%). These results are consistent with other published data during the pandemic. At 27 days versus 14 days, the total time on ECMO was almost twice as long in the COVID-19 patients. Despite the significantly longer time spent on mechanical support, the COVID-19 patients received fewer per-day RBC, FFP, and platelet transfusions, and only required more cryoprecipitate.

### 4.1. Strengths

We conducted this study at two major ECMO institutions in the United Kingdom. The collection of 271 patients is the first and largest of its kind during the pandemic reviewing transfusion requirements on VV-ECMO. Transfusion triggers were the same in both institutions following ELSO guidelines.

### 4.2. Limitations

Data were collected for the study retrospectively. There is potential for bias in data collection, data entry, and data quality assurance. The transfusion data were collected directly from the transfusion laboratories in both institutions. The patient demographics, however, were collected through the respective electronic databases and are thus dependent on accurate entry by staff.

The COVID-19 patients were a homogenous group with all the patients on VV-ECMO suffering from severe ARDS as a consequence of the same underlying respiratory pathology. The non-COVID patients are necessarily a more heterogenous group. The multiple disease processes seen in the non-COVID-19 group may have different transfusion requirements. Both RPH and USMH are part of the same national ECMO referral process. All included patients were accepted through this pathway, and we included only infectious pathologies in this study. No patients with trauma or chemical lung injury were included.

We did not collect data on the use of potentially thrombocytopenia-inducing drugs, particularly, furosemide and antibiotics. Both these are an integral part of the treatment of infective respiratory failure, and it is reasonable to assume that all included patients were exposed to them.

The mortality in non-COVID-19 patients was lower than that of those with COVID-19, providing for a relatively small number of deaths in the former group, putting these data at risk of being skewed.

### 4.3. Future Research

We accept that this retrospective cohort study poses more questions than it answers. The complex haematological processes occurring with severe COVID-19 are not fully understood. Further review of transfusion differences between COVID-19 waves or different COVID-19 strains may provide data on whether variations of the disease may contribute to the overall changes in coagulation. Transfusion obviously plays a big role in outcomes after severe respiratory illness requiring extracorporeal support. Particularly the association between mortality and platelet transfusion in COVID-19 patients and RBC, FFP, and cryoprecipitate transfusion in non-COVID-19 patients warrants further exploration. The impact of liberal versus restrictive transfusion practice on outcomes after mechanical extracorporeal respiratory support remains poorly understood.

## 5. Conclusions

This retrospective review of COVID-19 and non-COVID-19 patients requiring VV ECMO for respiratory failure demonstrates distinctly different transfusion patterns between the two groups of patients. Transfusion practice during mechanical extracorporeal respiratory support warrants further investigation and may shed further light on the haematological disease process in the case of COVID-19.

## Figures and Tables

**Table 1 jcm-12-04667-t001:** Baseline demographics.

Variable	COVID-19 (*n* = 168)		Non-COVID-19 (*n* = 103)		
	Median	95% CI	Median	95% CI	*p*
Age (year)	44.000	42.816–45.184	45.000	43.000–46.923	0.992
BMI (kg m^−2^)	30.000	28.399–31.062	28.100	26.250–30.915	0.036
Height (cm)	170.000	170.000–172.646	170.000	169.626–174.374	0.796
Weight (kg)	90.000	88.816–95.000	81.500	80.000–89.861	0.05
Male sex *n* (%)	105 (62.5%)		55 (53.4%)		
Haemoglobin (g L^−1^)	100.000	96.000–102.000	94.000	91.000–97.000	0.0282
Platelets (×10^9^ L^−1^)	243	222.816–269.184	168.000	149.077–194.923	<0.0001
Fibrinogen (g L^−1^)	5.290	4.704–5.700	4.380	3.365–5.263	0.0071

**Table 2 jcm-12-04667-t002:** Transfusion of blood products.

	COVID (*n* = 168)		Non-COVID (*n* = 103)		
Transfusion in Total					
	Mean	SD	Mean	SD	*p*
Red Blood Cells (total)	12.143	11.0564	12.515	15.3271	0.171
Platelets (total)	3.375	4.7773	4.728	9.7057	0.875
Fresh Frozen Plasma (total)	3.423	9.3453	5.010	12.7160	0.967
Cryoprecipitate (total)	3.292	6.4301	1.155	2.7751	0.022
Transfusion Indexed per Day					
Red Blood Cells (per day)	0.408	0.5123	0.996	2.6777	0.0005
Platelets (per day)	0.111	0.1639	0.296	0.6380	0.363
Fresh Frozen Plasma (per day)	0.101	0.3157	0.290	0.8003	0.976
Cryoprecipitate (per day)	0.117	0.2394	0.106	0.4831	0.022

**Table 3 jcm-12-04667-t003:** Transfusion of blood products indexed per day in survivors and deaths per group.

COVID-19	Alive (*n* = 88)		Dead (*n* = 80)		
	Mean	SD	Mean	SD	*p*
RBC per day (units/day)	0.347	0.2895	0.475	0.6737	0.187
Platelets per day (units/day)	0.0823	0.1342	0.142	0.1874	0.007
FFP per day (units/day)	0.118	0.3824	0.0820	0.2213	0.806
Cryoprecipitate per day	0.142	0.2556	0.0884	0.2183	0.189
**Non-COVID-19**	**Alive (*n* = 82)**		**Dead (*n* = 21)**		
RBC per day (units/day)	0.911	2.8479	1.331	1.8897	0.0008
Platelets per day (units/day)	0.251	0.6235	0.475	0.6777	0.118
FFP per day (units/day)	0.235	0.7818	0.504	0.8546	0.004
Cryoprecipitate per day (units/day)	0.0980	0.5279	0.136	0.2457	0.001

**Table 4 jcm-12-04667-t004:** Transfusion of blood products per day compared by group.

	COVID (*n* = 80)		Non-COVID (*n* = 21)		
Alive	Mean	SD	Mean	SD	*p*
RBC per day (units/day)	0.475	0.6737	1.331	1.8897	0.0001
Platelets per day (units/day)	0.142	0.1874	0.475	0.6777	0.191
FFP per day (units/day)	0.0820	0.2213	0.504	0.8546	0.011
Cryoprecipitate per day (units/day)	0.0884	0.2183	0.136	0.2457	0.243
**Dead**	**COVID (*n* = 80)**		**Non-COVID (*n* = 21)**		
RBC per day (units/day)	0.475	0.6737	1.331	1.8897	0.0001
Platelets per day (units/day)	0.142	0.1874	0.475	0.6777	0.191
FFP per day (units/day)	0.0820	0.2213	0.504	0.8546	0.011
Cryoprecipitate per day (units/day)	0.0884	0.2183	0.136	0.2457	0.243

**Table 5 jcm-12-04667-t005:** Mortality and days spent on ECMO during the study period.

	COVID-19 (*n* = 168)		Non-COVID-19 (*n* = 103)		*p*
Days on ECMO (Median, 95% CI)	27.1	21.00–33.18	14.16	11.67–15.66	<0.0001
Mortality (%)	80 (47.6)		21 (20.4)		0.0001

## Data Availability

Patient data used in this study are held in secure hospital data storage and can be made available in deidentified form if requested and after applying for the appropriate institutional permissions.

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
