# Peer review of "Comparison of Blood and Blood Product Transfusion in COVID-19 and Non-COVID-19 Patients Requiring Extracorporeal Membrane Oxygenation for Severe Respiratory Failure"

_jcm, 2023, doi:10.3390/jcm12144667_

Round 1

Reviewer 1 Report

I read with great interest the manuscript by Fernando et al. on the comparison of blood transfusion between COVID-19 and non-COVID-19 patients in ECMO for respiratory failure. The paper is sound and well written. I only have some comments:

- Line 38-39. Authors should add the new technologies and the use of new anticoagulants as possible explanations for the lower incidence of bleeding in ECMO patients (doi: 10.1097/CCM.0000000000005033 - doi: 10.1111/aor.14276). Please briefly discuss and add these 2 references.

- Why authors did not exclude anemic patients or patients with ematological malignancies? Please explain.

- Line 90, 111, Table 1. There is no need to specify the hospital names since the protocol was applied to both of them equally.

- Results. Please present the primary outcomes before the secondary outcomes.

- There are too many tables in the result section. I would suggest to reduce the number by unifying them.

- Line 213, 214, 217. Please be consistent with the name of COVID-19 through the paper.

Author Response

I read with great interest the manuscript by Fernando et al. on the comparison of blood transfusion between COVID-19 and non-COVID-19 patients in ECMO for respiratory failure. The paper is sound and well written. I only have some comments:

Comment 1: Line 38-39. Authors should add the new technologies and the use of new anticoagulants as possible explanations for the lower incidence of bleeding in ECMO patients (doi: 10.1097/CCM.0000000000005033 - doi: 10.1111/aor.14276). Please briefly discuss and add these 2 references.

Response 1: Thank you for this valuable comment. We have added the following to the Introduction: ‘Although unfractionated heparin remains the most widely used anticoagulant during ECMO, there have been recent advances using other agents, namely bivalirudin and mafamostat mesilate [8,9]. One large case series with central mostly veno-arterial ECMO describes a reduction in transfusion in the first 24 hours after ECMO initiation and reduced mortality in the bivalirudin patients compared to those on heparin [8]. These findings were corroborated in later another single-centre series including veno-venous ECMO patients, where patients on bivalirudin received a significantly lower number of blood and blood products and experienced fewer thrombotic complications [10].

We have added the references the reviewer suggested [8, 9]. We have also taken the liberty to add a third reference pertaining to the use to bivalirudin in VV ECMO [10], seeing that [8] mostly includes VA ECMO patients.

Comment 2: Why authors did not exclude anemic patients or patients with ematological malignancies? Please explain.

Response 2: Thank you for this question which clearly highlights a need for clarification. VV ECMO in the UK is a nationally commissioned service with clear admission criteria all 5 ECMO centres adhere to. Any sort of malignancy – haematological or solid organ – is a clear contraindication for accepting a patient for ECMO treatment. Being very much ensconced in this system, it had not occurred to us that this requires an extra mention.  We have added the following sentence and a supporting reference to the Methods section: ‘Both centres adhered to the UK nationally agreed exclusion criteria for respiratory ECMO, which include significant life-limiting comorbidity such as haematological or solid organ malignancy [16].
Had we applied the WHO criteria for anaemia (Hb < 120g/l in women and < 130g/l in men) we would not have been left with many patients to include, seeing that the median Hb was 100 g/l in the Covid-19 and 94 in the non-Covid-19 group. In our practice we almost never see critically ill patients requiring ECMO with normal Hb. We hope the reviewer accepts that it is more the homogeneity of the groups rather than the value of one parameter that provides the strength of the present study.

Comment 3: Line 90, 111, Table 1. There is no need to specify the hospital names since the protocol was applied to both of them equally.

Response 3: Thank you for pointing this out. We completely agree and have removed the hospital names.

Comment 4: Results. Please present the primary outcomes before the secondary outcomes.

Response 4: Again, a fair point and well taken. We have moved the outcomes to the end to the Results section. It is now structured in i) baseline and admission characteristics, ii) transfusion and iii) outcomes. 

Comment 5: There are too many tables in the result section. I would suggest to reduce the number by unifying them.

Response 5: Thank you for this suggestion, which does not come as much of a surprise. We have unified the tables, cutting the number down to 5. Contrary to our initial fears, this does not seem to make interpreting them any more difficult and we are happy with the result.

Comment 6: Line 213, 214, 217. Please be consistent with the name of COVID-19 through the paper.

Response 6: Thank you for pointing out these inconsistencies, which were present throughout the paper. The spelling is now COVID all the way. 

Reviewer 2 Report

Content suggestions:

1.         Could the authors classify the patients or add any information about the comorbidities and drugs used by the patients ?

2.         I would like to kindly ask the authors to do the same – to add more data about the drugs taken by the patients that might induce thrombocytopenia ? I do not suppose that they developed thrombocytopenia, I ask for this just for the completness.

3.         Which are the indication criteria of cryoprecipitate in these patients ? When do the authors administer FFP and when cryoprecipitate ?

4.         Can the authors reveal the statistics of fibrinogen administration in the patients ? It is also commonly used to correct coagulopathy...

I assess this manuscript as very important for the clinical practice, as I sincerely appreciate each new information about this life-threatening infection. 

Author Response

Comment 1: I assess this manuscript as very important for the clinical practice, as I sincerely appreciate each new information about this life-threatening infection.

Response 1: Thank you! We very much appreciate this comment. Although it is very clinical in nature, we hope our study will generate further investigation into the complexities surrounding COVID-19 and coagulation.

Comment 2: Could the authors classify the patients or add any information about the comorbidities and drugs used by the patients ?

Response 2: Thank you for this comment. We have added the following sentence to the Methods section: ‘Both centres adhered to the UK nationally agreed exclusion criteria for respiratory ECMO, which include significant life-limiting comorbidity such as haematological or solid organ malignancy [16].’ We hope this clarifies the position on mostly exclusion criteria, which are very stringent in the UK. Patients basically need to have respiratory failure of not-cardiac origin and be in single organ failure to be admitted for ECMO at one of the five referral centres in the country. For UK purposes, co-morbidities are not relevant due to the highly selective nature of this nationally driven – but also controlled – service. May we point the reviewer to the following in the Limitations section: Both RPH and USMH are part of the same national ECMO referral process. All included patients were accepted through this pathway and we included only infectious pathologies in this study. No patients with trauma or chemical lung injury were included. We hope this goes some way to alleviate their concerns.
The reviewer’s question about drugs is less straight forward. Like the majority of patients in respiratory failure of infective origin, those included in our cohort have been administered drugs like furosemide and various antibiotics. All these drugs are known to have the potential to cause thrombocytopenia. As particularly furosemide and antibiotics are an essential part of treating severe respiratory failure and bacterial superinfections, we did not collect this data. We do not consider this data to be essential as all patients will have been treated the same way. We feel that including it has the risk of making the paper messy with more tables and detract from the main message. We understand the reviewer’s concerns and have added the following sentence to the Limitations section: ‘We did not collect data on the use of potentially thrombocytopenia-inducing drugs, particularly furosemide and and antibiotics. Both these are an integral part of the treatment of infective respiratory failure and it is reasonable to assume that all included patients were exposed to them.’ We hope that this is acceptable to this reviewer.

Comment 3: I would like to kindly ask the authors to do the same – to add more data about the drugs taken by the patients that might induce thrombocytopenia ? I do not suppose that they developed thrombocytopenia, I ask for this just for the completness.

Response 3: Thank you for this comment. We believe that this has already been addressed in Response 2.

Comment 4: Which are the indication criteria of cryoprecipitate in these patients ? When do the authors administer FFP and when cryoprecipitate ?

Response 4: The transfusion approach is laboratory and point of care based. Cryoprecipitate is only given when fibrinogen is below the transfusion threshold; FFP is given when aPTT and / or PT are deranged. We refer the reviewer to the Transfusion Management section of the Methods, where our practice is explained.

Comment 5: Can the authors reveal the statistics of fibrinogen administration in the patients ? It is also commonly used to correct coagulopathy...

Response 5: Thank you for this comment. We are assuming that the reviewer is referring to the administration of fibrinogen concentrate? In answer to this question, we can confirm that no patient included in the analysis received either fibrinogen or factor concentrates.

Reviewer 3 Report

Major

  Research has shown that while concerns over blood transfusion-transmission risk is legitimate, SARS-CoV-2 itself has no direct threat to blood safety. Viral RNA in asymptomatic donors, who provide the majority of donations, is extremely low even when detected.

  It has been reported that COVID-19 has resulted in an exponential increase in patients with severe respiratory failure due to ARDS requiring extracorporeal membrane oxygenation (ECMO). Patients on ECMO regularly require high volumes of blood and blood products, but so far there has been no comparison of transfusion requirements between COVID-19 and non-COVID-19.

The current study examines the transfusion requirements of patients requiring ECMO. Since COVID-19 patients spent almost twice as long on ECMO (27.1 vs 14.16 days, p = <0.0001) we indexed transfusion in both groups to days on ECMO to allow comparison. COVID-19 patients required less red blood cells (RBC) per day (0.408 vs 0.996, p = 0.0005) but more cryoprecipitate transfusions (0.117 vs 0.106, p = 0.022) compared to non-COVID-19 patients. Covid-19 patients had more than double the mortality of non-Covid-19 patients and those who died during the study period had higher platelet transfusion requirements then their non-Covid-19 counterparts. Transfusion requirements and coagulopathy differ between COVID-19 and non-COVID-19 patients.

These observations are very important the future ECMO management of unknown pandemic virus. However, as authors have stated that the distinctly different transfusion patterns between the two groups remain difficult to interpret.

The inconclusive study unfortunately has given us the useful information about blood product transfusion in ECMO treatment due to COVID-19 like viruses.

Author Response

Comment 1: Research has shown that while concerns over blood transfusion-transmission risk is legitimate, SARS-CoV-2 itself has no direct threat to blood safety. Viral RNA in asymptomatic donors, who provide the majority of donations, is extremely low even when detected.

Response 1: Thank you for this comment, which has educated us. As the present paper is not investigating blood donation after surviving VV ECMO with or without COVID-19, but transfusion during ECMO, we are unsure what to do with this comment and can’t really see a place for it in the manuscript.

Comment 2: It has been reported that COVID-19 has resulted in an exponential increase in patients with severe respiratory failure due to ARDS requiring extracorporeal membrane oxygenation (ECMO). Patients on ECMO regularly require high volumes of blood and blood products, but so far there has been no comparison of transfusion requirements between COVID-19 and non-COVID-19. 

The current study examines the transfusion requirements of patients requiring ECMO. Since COVID-19 patients spent almost twice as long on ECMO (27.1 vs 14.16 days, p = <0.0001) we indexed transfusion in both groups to days on ECMO to allow comparison. COVID-19 patients required less red blood cells (RBC) per day (0.408 vs 0.996, p = 0.0005) but more cryoprecipitate transfusions (0.117 vs 0.106, p = 0.022) compared to non-COVID-19 patients. Covid-19 patients had more than double the mortality of non-Covid-19 patients and those who died during the study period had higher platelet transfusion requirements then their non-Covid-19 counterparts. Transfusion requirements and coagulopathy differ between COVID-19 and non-COVID-19 patients. 

These observations are very important the future ECMO management of unknown pandemic virus. However, as authors have stated that the distinctly different transfusion patterns between the two groups remain difficult to interpret.

The inconclusive study unfortunately has given us the useful information about blood product transfusion in ECMO treatment due to COVID-19 like viruses.

Response 2: Thank you for this comment, which largely is a repetition of the abstract. We maintain that much remains to be learned about the blood and coagulation issues surrounding COVID-19, but also those surrounding extracorporeal circulation in non-COVID-19 critical illness. Our study is observational and not basic science in nature and is not claiming to explain the phenomenon of different transfusion patterns between patients on ECMO for COVID-19 compared to those on ECMO for other infections.
It is meant to be hypothesis-generating. Does that necessarily make it inconclusive? We would argue the opposite is true as we demonstrate distinctly different patterns in transfusion between the two disease aetiologies.
We apologise for not being able to ascertain what this reviewer is suggesting we change in our manuscript and have not altered anything as a result of the above comments.

Round 2

Reviewer 3 Report

Well revised